# Revolutionizing Breast Reconstruction: The Rise of Hybrid Techniques

**DOI:** 10.3390/medicina61081434

**Published:** 2025-08-09

**Authors:** Evan Rothchild, Isabelle T. Smith, Gabrielle Odoom, Mark L. Smith, Neil Tanna

**Affiliations:** Division of Plastic and Reconstructive Surgery, Northwell Health, Great Neck, NY 11021, USA; evan.rothchild@gmail.com (E.R.); isabelletsmith@gmail.com (I.T.S.); gabbyodoom9@gmail.com (G.O.); msmith63@northwell.edu (M.L.S.)

**Keywords:** hybrid breast reconstruction, autologous reconstruction, DIEP flap, breast implant, acellular dermal matrix

## Abstract

Hybrid breast reconstruction (HBR) combines autologous tissue and bio-prosthetic breast reconstruction techniques. This method addresses many challenges associated with stand-alone techniques, including inadequate volume with autologous reconstruction and esthetic issues like rippling in implant-based reconstruction. However, despite its promising advantages, HBR remains underutilized. This review explores the development of HBR, surgical techniques, clinical outcomes, current barriers to adoption, and the future potential of this innovative approach to breast reconstructive surgery.

## 1. Introduction

The demand for post-mastectomy breast reconstruction has accelerated dramatically, with procedure rates increasing 75% since 2000, reflecting both expanded insurance coverage and growing recognition of its psychological benefits [1,2]. As reconstructive options diversify, patients face increasingly complex decisions about which approach best aligns with their anatomical constraints and esthetic goals. The reconstructive landscape has traditionally been dominated by a dichotomous choice between prosthetic devices and autologous tissue transfer, with each option offering distinct advantages and limitations [3].

Implant-based reconstruction remains the most common approach, accounting for approximately 80% of procedures in the United States [4]. Its popularity stems from technical simplicity, widespread surgical availability, and lack of donor site morbidity [5]. However, implants can introduce significant concerns, including implant palpability, visibility, rippling, rupture, need for surveillance, and capsular contracture affecting up to 30% of recipients within 10 years [6]. Additionally, implant-based reconstruction carries higher complication rates in irradiated fields [7].

In contrast, autologous reconstruction utilizes the patient’s own tissue, which provides several advantages over implants. The use of vascularized flaps results in softer, more natural-feeling breasts that can fluctuate with the patient’s weight [8,9]. Furthermore, autologous reconstruction avoids implant-related complications and the need for future revisions [8]. Studies consistently demonstrate superior patient-reported outcomes regarding satisfaction and psychosocial functioning with autologous techniques compared to implant-based approaches [10,11]. However, these procedures are more technically complex and introduce potential donor site morbidity [12,13,14]. The evolution from pedicled Transverse Rectus Abdominis Musculocutaneous (TRAM) flap to perforator-based techniques like the Deep Inferior Epigastric Perforator free-flap (DIEP) has reduced abdominal wall morbidity while preserving key benefits.

Although autologous reconstruction is gaining preference for many surgeons, a fundamental challenge of autologous breast reconstruction arises in a specific subset of patients: the mismatch between desired breast volume and available donor tissue. This volume discordance particularly affects thin women with minimal abdominal tissue, patients desiring larger breast reconstruction, women with previous abdominal surgeries, and women requiring bilateral reconstruction [15,16,17]. When a volume discrepancy exists, surgeons and patients face difficult decisions that often require compromising either esthetic outcomes or volumetric goals. Potential options to address this challenge include fat grafting and stacked flaps. Fat grafting has been used as a secondary measure to add volume or fix deformities following reconstruction [18,19,20,21,22]. However, fat grafting has limitations, such as variable fat retention and fat necrosis, which can impact the final esthetic result [20,23]. Stacked flaps, which involve using multiple flaps to achieve the desired volume, are another option [24]. While this approach can help achieve the desired volume, it is associated with increased technical complexity, longer operative times, and potentially higher donor site morbidity [25,26,27].

As an alternative to these techniques, hybrid breast reconstruction (HBR) has emerged as a promising solution to address the challenge of volume discordance. HBR combines autologous tissue with implants or acellular dermal matrix (ADM) to achieve the desired breast volume while maintaining the benefits of autologous reconstruction [28]. The key concept underlying all variations in hybrid techniques is the synergistic combination of autologous tissue with alloplastic materials or ADM. The flap provides vascularized soft tissue coverage, which enhances the natural feel of the reconstructed breast and minimizes complications associated with implants or ADM. The underlying implant or ADM provides additional core projection and volume, enabling patients with limited donor site tissue to achieve their desired breast size and shape.

The history of HBR spans several decades, with older techniques utilizing various flap types, such as the latissimus dorsi and TRAM flaps, often combined with subpectoral implant placement [16,29,30,31]. These early hybrid techniques laid the groundwork for modern approaches that aim to minimize morbidity and optimize outcomes. In 2018, Momeni and Kanchwala described a significant advancement in HBR, utilizing a DIEP flap combined with a prepectoral implant [32]. This marked a shift from the traditional subpectoral implant placement, which was associated with increased postoperative pain, animation deformity, and other complications [3,32,33]. Silverstein and Momeni (2023) further validated the long-term viability of the hybrid approach in a study with a 2-year follow-up, demonstrating that ADM-based HBR was associated with lower rates of capsular contracture and implant malposition [34]. Wang et al. (2024) conducted the largest comparative study to date, demonstrating that a hybrid DIEP group had a lower infection rate compared to an implant-only group while maintaining similar rates of other complications and flap success [35].

## 2. Methods

This narrative review will explore two novel HBR techniques described by the senior authors (NT, MS): HyFIL^®^ (Hybrid Flap, Implant, Lipofilling) and HyPAD^®^ (Hybrid Flap, Prepectoral Acellular Dermal Matrix (Figure 1) [28,36]. Specifically, the review will provide an overview of the clinical indications, operative workflows, surgical techniques, and patient outcomes associated with each HBR approach. Additionally, an illustrative clinical case performed by the senior authors (NT, MS) will be presented for each technique.

## 3. HyFIL^®^

### 3.1. Description of Technique

The HyFIL^®^ technique–Hybrid Flap, Implant, Lipofilling–integrates a free flap with a prepectoral implant and optional fat grafting to achieve natural contour, volume, and projection. A small silicone or saline round implant is circumferentially wrapped with ADM and positioned in the prepectoral plane. The implant-ADM construct is secured to the anterior chest wall, providing structural support and controlling implant position while preserving access to the internal mammary vessels for subsequent microsurgical anastomosis. The flap is then inset over the implant-ADM construct. Together, the implant enhances core projection and volume while the overlying vascularized flap offers soft-tissue camouflage and natural contour (Appendix A). Lipofilling (fat grafting) can then be used at a later date to improve the appearance and contour of the breasts as needed.

### 3.2. Patient Selection for HyFIL^®^

HyFIL^®^ is optimally suited for patients with a substantial mismatch between available donor flap volume and desired breast volume–particularly when this discrepancy exceeds the augmentation possible with ADM alone. It is especially advantageous for thin patients with minimal abdominal adiposity and/or laxity, those pursuing bilateral reconstruction, or patients seeking a larger breast size and more robust core projection. Unlike the HyPAD^®^ technique, which uses stacked ADM to modestly augment flap volume, HyFIL^®^ allows for significantly greater volume augmentation. Implant volumes in this setting typically range from 120 to 250 cc, whereas even the thickest folded ADM constructs in the HyPAD^®^ technique provide only about 75 to 140 cc of augmentation.

### 3.3. Case 1—HyFIL^®^

The authors report on a 37-year-old female who presented with invasive ductal carcinoma of the left breast. The patient initially underwent left breast lumpectomy and bilateral oncoplastic breast reduction. She then decided to undergo bilateral nipple-areolar sparing mastectomies. During consultation, the patient expressed a strong preference for simultaneous autologous breast reconstruction utilizing a deep inferior epigastric artery perforator (DIEP) flap. However, her abdominal donor site tissue was insufficient to achieve her desired breast reconstruction volume. After a comprehensive discussion regarding reconstruction methods, including autologous and implant-based approaches, she was specifically counseled regarding an alternative option based on her reconstructive goals: HBR combining DIEP flaps with acellular dermal matrix (ADM) and/or prepectoral implants.

It was explained that ADM would be required if implants were utilized, although ADM could also be placed without implants. The patient was specifically informed that ADM use in breast reconstruction is considered an off-label indication by the FDA and carries its own set of risks and potential complications. Implant types, particularly smooth round silicone implants, were discussed extensively. The patient understood that HBR could be performed immediately at the time of mastectomy or alternatively pursued as a delayed procedure with implant and/or ADM insertion at a later date. She further acknowledged that HBR carries risks associated with both flap-based and implant-based techniques. A preoperative magnetic resonance angiogram (MRA) of the abdomen and pelvis was performed to identify abdominal perforators and facilitate surgical planning.

### 3.4. Operative Technique

The patient was marked in the preoperative holding area in the upright position. The vertical midline of the chest and abdomen was marked. The breasts were outlined bilaterally, and the vertical meridian was marked. A circumvertical mastectomy incision was utilized as the patient had a previous breast reduction. The superior aspect of the abdominal ellipse was marked at the level of the umbilicus, and the inferior aspect was marked within a natural suprapubic crease. The breast surgeon performed bilateral nipple-areolar sparing mastectomies in standard fashion (580 g left, 640 g right). Upon completion, the breast reconstruction sites were copiously irrigated with an antibiotic solution.

For breast reconstruction, bilateral 155 cc Sientra smooth, round, low-profile silicone implants were selected based on patient preference, mastectomy specimen weights, and hemi-abdominal flap weight. One sheet of ADM was used for each implant. Each ADM was prepared and rinsed according to the manufacturer’s instructions, wrapped circumferentially around its corresponding implant, and secured with polydioxanone suture (Figure 2 and Figure 3 and Appendix A). Intraoperative images demonstrate the added projection and volume resulting from combining the flap with the implant-ADM construct (Figure 4). The implant-ADM constructs (195 g left, 205 g right) were secured to the prepectoral chest wall with circumferential polydioxanone sutures (Figure 5).

The third intercostal space was identified, and the pectoralis major muscle was split parallel to its fibers. The perichondrium of the third and fourth ribs was incised, and subperichondrial dissection was performed. Using a rongeur, the fourth rib was removed. The posterior perichondrium and intervening intercostal muscles were removed from lateral to medial. This exposed the internal mammary artery and vein, which were circumferentially freed for recipient site microsurgery (Figure 6).

The abdominal donor site was dissected as per the planned design, with preservation of vascular perforators. The hemi-abdominal flaps (515 g left, 520 g right) were elevated, and dominant perforators were traced and preserved. The deep inferior epigastric artery and vein were dissected and traced to their origin at the external iliac vessels. Each flap was transferred contralaterally to its respective breast reconstruction site for microsurgical anastomosis, with arterial anastomosis performed under an operating microscope using 9-0 nylon sutures and venous anastomosis performed using a venous coupler. Pulsatile flow, good color, and capillary refill were confirmed. Each flap was inset circumferentially, fully covering the implant-ADM construct (Figure 7). Mastectomy skin flaps were examined and showed no signs of ischemia. A small skin paddle from each flap was inset along the incision line for postoperative monitoring, and incisions were closed in layers using 3-0 Monocryl for the deep dermal layer, followed by subcuticular closure.

Bilateral 19-French closed-suction drains were placed at both the breast and abdominal sites through separate stab incisions and secured with silk sutures. The abdominal donor site was irrigated, hemostasis was ensured, and a subfascial Phasix mesh was secured with horizontal mattress polydioxanone sutures. The anterior rectus fascia was reinforced with polydioxanone barbed suture. Abdominal closure and umbilical transposition were performed with absorbable monofilament sutures. The patient tolerated the procedure well, with no intraoperative complications noted.

## 4. HyPAD^®^

### 4.1. Description of Technique

The HyPAD^®^ technique–Hybrid Flap, Prepectoral Acellular Dermal Matrix–combines a free flap with a stacked ADM construct to augment breast volume and core projection without the use of an implant. Acellular dermal matrix is folded and layered into a compact, high-density configuration. This ADM construct is then secured to the anterior chest wall in the prepectoral plane, ensuring preservation of access to recipient vessels. The flap is subsequently inset over the ADM, providing full vascularized coverage and soft-tissue integration (Figure 8). The result is a modest yet meaningful enhancement in projection and volume, with the added benefit of maintaining an implant-free reconstruction (Appendix A).

### 4.2. Patient Selection for HyPAD^®^

HyPAD^®^ is best suited for patients who desire autologous reconstruction but have a mild-to-moderate volume deficit relative to their reconstructive goals. Compared to HyFIL^®^, HyPAD^®^ offers more limited volumetric enhancement–typically in the range of 75–140 cc. Patients with thinner body habitus, small volume discrepancies, or esthetic goals focused more on natural contour than maximal projection may benefit most from this technique. HyPAD^®^ is also particularly appropriate for those who wish to avoid implants due to personal preference or concerns regarding long-term surveillance or device complications.

### 4.3. Case 2—HyPAD^®^

The authors report on a 47-year-old female who presented to the clinic to discuss her reconstructive options in anticipation of a bilateral prophylactic nipple-sparing mastectomies. During consultation, we reviewed a range of reconstructive strategies (including foregoing reconstruction, implant-based, and autologous techniques) and discussed the timing options (immediate versus delayed reconstruction). Although several approaches were considered, she expressed a clear preference for autologous reconstruction and specifically elected to pursue deep inferior epigastric perforator (DIEP) flap reconstruction.

Given that the patient’s abdominal tissue is smaller than her current breast volume, the potential need for ADM became an important consideration. We explained that if the DIEP flap does not provide sufficient core projection, supplementation with acellular dermal matrix (ADM) could be an option. However, we specifically noted that its use in breast reconstruction is considered off-label by the FDA. After a comprehensive discussion of these options and their respective risks and benefits, the patient provided informed consent to proceed with the proposed surgical plan. Preoperative MRA of the abdomen and pelvis was obtained.

### 4.4. Operative Technique

A bilateral nipple-areolar sparing mastectomy was performed by the breast surgeon in standard fashion. Recipient vessel preparation and abdominal flap harvest were performed similarly to case 1. According to the patient’s preference, mastectomy specimen weight, and hemiabdominal flap weight, the decision was made to augment the breast reconstruction volume (Figure 9). The acellular dermal matrix was rinsed according to the manufacturer’s instructions and sutured in a stacked fashion using 3-0 PDS sutures (Appendix A). The stacked ADM construct was then secured to the chest wall using additional 3-0 PDS sutures (Figure 10 and Figure 11).

The flaps were transferred to the recipient site in succession (Figure 12). Utilizing an operating microscope, microsurgical anastomosis was performed, similar to case 1. The flaps were inset circumferentially using polydioxanone sutures, ensuring that the entire stacked acellular dermal matrix construct was completely covered. Breast and abdominal closure were performed similar to case 1.

## 5. Discussion

HBR has become increasingly popular over the past decade. Integrating a free flap with either a prosthetic implant or ADM combines the strengths of both traditional methods: the flap provides well-vascularized soft tissue for a natural contour and ptosis, while the implant or ADM adds core volume and projection [3,37]. It is a valuable solution in a volumetric mismatch between available donor tissue and the desired breast reconstruction volume [32]. This is particularly relevant for patients with low BMI, prior abdominal surgery, macromastia, or ptotic breasts [38]. Even flap-to-implant ratios as low as 1:5 have provided significant cosmetic benefits by improving central breast projection and enabling the flap to adequately camouflage the underlying implant [31].

The HBR approach offers many benefits beyond esthetics. HBR reduces the need for stacked flaps or harvesting from alternative donor sites, which may reduce operative morbidity [3]. Furthermore, in contrast to implant-based reconstruction alone, hybrid reconstructions respond to weight changes over time and follow the natural curvature of the breast [38]. Having flap coverage for an implant has also been shown to reduce the risk of capsular contracture, implant exposure, and wound breakdown [3].

The protective effect of vascularized flap tissue has been shown to benefit patients. In irradiated breasts, HBR demonstrates lower rates of complications and better cosmesis due to the protective flap layer [15]. In adjunctive procedures such as fat grafting and nipple reconstruction, the well-vascularized tissue overlying the implant enhances fat graft take and the durability of nipple projection [3,39]. In cases of mastectomy skin flap necrosis, HBR patients tend to have more favorable wound healing outcomes and a reduced need for additional surgeries [32]. Furthermore, a vascularized flap over an implant may offer a protective benefit against infection, likely due to increased perfusion and tissue oxygenation, promoting microbial resistance and wound healing [35].

For DIEP flap patients, HBR can contribute to superior donor-site cosmesis in select cases because alloplastic volume supplementation reduces the volume required from the flap. Consequently, the abdominal harvest in a DIEP flap can result in a lower, more esthetic scar than full-volume DIEP reconstructions [3,32]. The resulting scar can be more similar to a cosmetic abdominoplasty or “tummy tuck” scar rather than the higher and less esthetically pleasing scar often required to harvest enough volume for breast reconstruction [3,32].

A significant concern with combining autologous and implant-based approaches in HBR is whether the risk of postoperative complications is additive. The major complications following HBR have been reported to be fat necrosis, flap loss, venous congestion, and mastectomy skin necrosis [3]. However, studies have found that postoperative complications from HBR occur at similar or lower rates than with autologous or implant-based reconstruction alone [3]. Notably, non-implant-related complications in HBR tend to be similar to those seen in flap-only techniques [3]. In contrast, implant-specific issues such as palpability, rippling, capsular contracture, and reconstructive failures tend to occur at lower rates than in implant-only reconstruction [15,30,40]. Including ADM has been shown to further reduce implant-related complication rates in HBR [41].

Another primary concern with HBR is the increased cost compared to flap-only or implant-only techniques. Particularly in cases where ADM is utilized, the upfront price difference can be substantial. While the initial cost is higher, fewer implant complications may reduce the need for secondary procedures, ultimately offsetting this expense over time [42,43,44]. Still, the financial burden remains a considerable barrier in some settings, mainly when using ADM.

Each technique described in this review presents its own set of practical considerations. In HyFIL^®^, the use of implants requires patients to adhere to FDA-recommended screening and imaging protocols, increasing healthcare costs and time demands. Specifically, the FDA recommends MRI screening for silicone implants 5 years after placement and every 2 to 3 years afterward to detect silent rupture. These screenings, along with surgical interventions for implant rupture or expiration, contributed to over $33 million in healthcare costs in 2010 alone, with costs continuing to rise [45,46,47]. Patients hoping to avoid implants and their associated long-term surveillance demands may opt for the HyPAD^®^ technique. However, the augmentation capacity of even the thickest ADM constructs is limited to approximately 90 to 140 cc, whereas implants in this context range from 120 to 250 cc. As a result, HyPAD^®^ may be less suitable for patients requiring substantial additional volume. Currently, no technique provides substantial alloplastic volume supplementation without the use of implants. Furthermore, the use of ADM in breast reconstruction remains off-label under existing FDA guidelines.

The distinct risk profiles of these techniques raise additional considerations. Both techniques incorporate a flap and ADM, but only HyFIL^®^ uses an implant. As a result, HyFIL^®^ carries added risks associated with implant-based reconstruction, including infection, explant, malposition, capsular contracture, and unnatural breast contours [6]. These risks are absent in the HyPAD^®^ approach. Therefore, while both techniques share complication risks related to flap and ADM use, the presence of an implant in HyFIL^®^ introduces additional considerations for patient counseling, surgical planning, and procedural outcomes. Further research on long-term complication rates for HyFIL^®^ and HyPAD^®^ is essential to more precisely compare their respective risk profiles.

Although HBR has shown great promise as a reconstructive strategy, there is currently a scarcity of level 1 evidence supporting these techniques. Most available studies are limited by small sample sizes, retrospective designs, heterogeneous flap types, and the absence of comparator groups, making it challenging to conclude outcomes relative to flap-only or implant-only approaches. Also, because these are novel techniques, there is a notable lack of long-term follow-up data on complication rates and patient-reported satisfaction. Future research should prioritize well-designed comparative studies that directly evaluate HBR techniques like HyFIL^®^ and HyPAD^®^ against traditional flap-only reconstructions. Moreover, cost–benefit analyses are critical because of the higher initial costs associated with hybrid procedures and ADM use.

## 6. Conclusions

HBR has emerged as a strong option for patients requiring breast reconstruction who have a discrepancy between their donor site flap volume and desired breast reconstruction volume [3,28,32]. Combining autologous flap reconstruction with a prepectoral implant and/or ADM, HBR allows patients to experience the benefits of both procedures. Specifically, the implant and/or ADM add core volume and projection, and the flap can offer natural contour and ptosis [3,37]. The HyFIL^®^ technique—Hybrid Flap, Implant, Lipofilling—integrates a free flap with a prepectoral implant and optional fat grafting to achieve natural contour, volume, and projection. This technique is primarily suited for patients with a substantial volume mismatch who require significant additional projection that cannot be achieved through ADM alone. The HyPAD^®^ technique—Hybrid Flap, Prepectoral Acellular Dermal Matrix—combines a free flap with a stacked ADM construct to augment breast volume and core projection without an implant. This technique is best for patients who desire implant-free reconstruction and have only a mild-to-moderate volume deficit. HBR techniques have shown promising early outcomes and have expanded the pool of candidates eligible for autologous breast reconstruction.

## Figures and Tables

**Figure 1 medicina-61-01434-f001:**
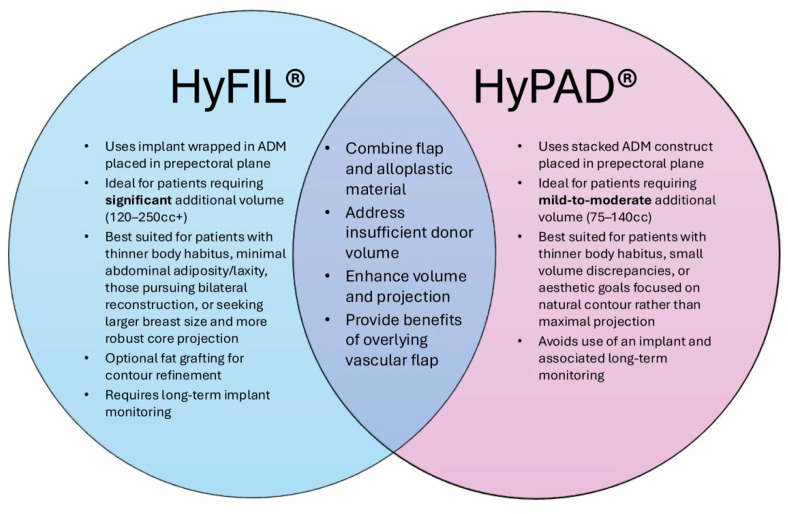
Venn diagram summarizing the key similarities and differences between HyFIL^®^ and HyPAD^®^ techniques in hybrid breast reconstruction.

**Figure 2 medicina-61-01434-f002:**
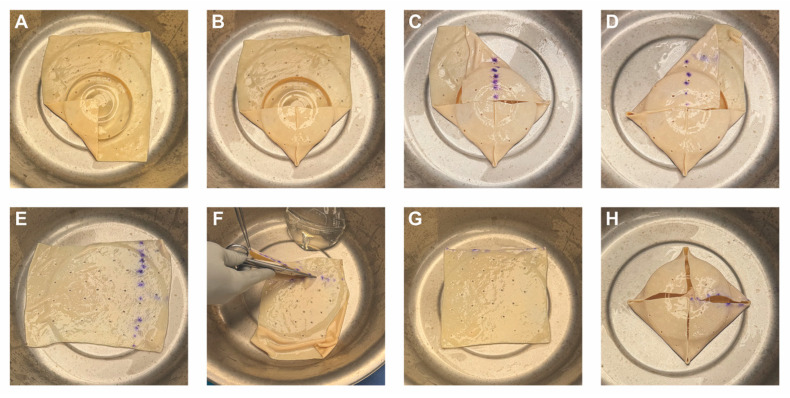
Stepwise demonstration of complete circumferential wrapping of a 155cc smooth, round, low-profile implant using a 16 × 20 cm sheet of acellular dermal matrix (ADM) in HyFIL^®^ Breast Reconstruction. (**A**,**B**) Two corners of the ADM are folded inward toward the center. (**C**,**D**) The remaining corners are folded in and excess ADM is marked for removal. (**E**) The markings indicate approximately 2 cm of ADM to be removed. (**F**) The excess ADM is cut off, converting the rectangular sheet of ADM into a square. (**G**) The square ADM is laid flat with the dermal side facing down. (**H**) A small, round, silicone implant is placed in the center of the square ADM with the anterior surface facing down. The four corners of the ADM are then brought together to be secured using 3-0 PDS suture.

**Figure 3 medicina-61-01434-f003:**
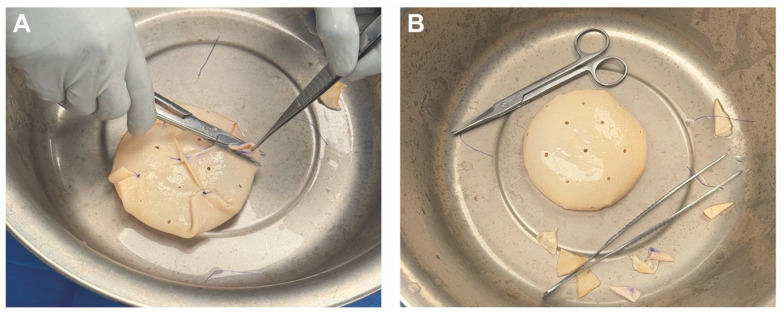
The implant-ADM construct is shown following completion of the wrapping technique. (**A**) Excess ADM is carefully trimmed using scissors to ensure a smooth surface without folds or irregularities. (**B**) The final implant-ADM construct with the implant fully enveloped in a single layer of ADM.

**Figure 4 medicina-61-01434-f004:**
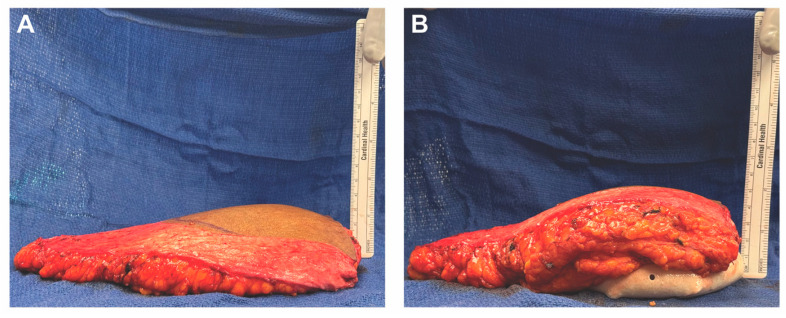
The implant provides additional volume and projection to the DIEP flap. (**A**) The DIEP flap alone had a projection of about 3.5 cm. (**B**) The DIEP flap combined with the implant-ADM construct had a projection of about 5 cm.

**Figure 5 medicina-61-01434-f005:**
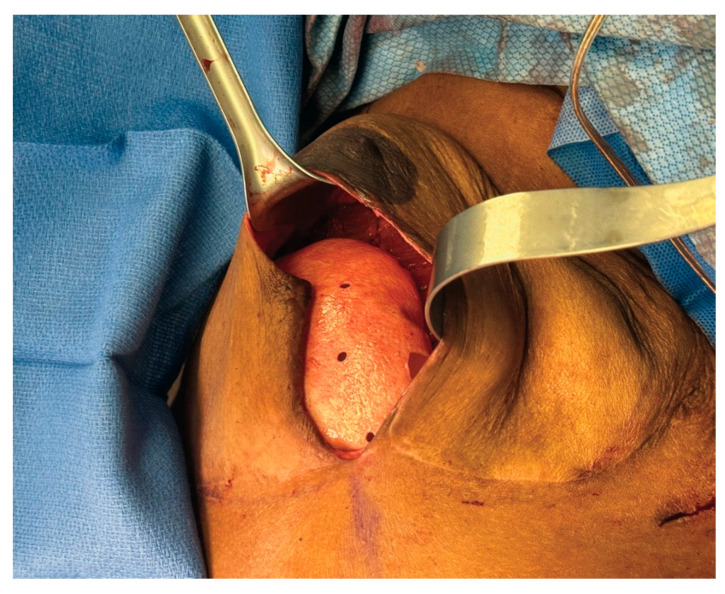
The implant-ADM construct is secured circumferentially to the left chest wall using 3-0 PDS sutures. The construct is positioned inferiorly, leaving adequate space for microsurgical anastomosis to the internal mammary vessels. In most cases, the implant-ADM construct can be placed prior to the flap anastomosis and placement.

**Figure 6 medicina-61-01434-f006:**
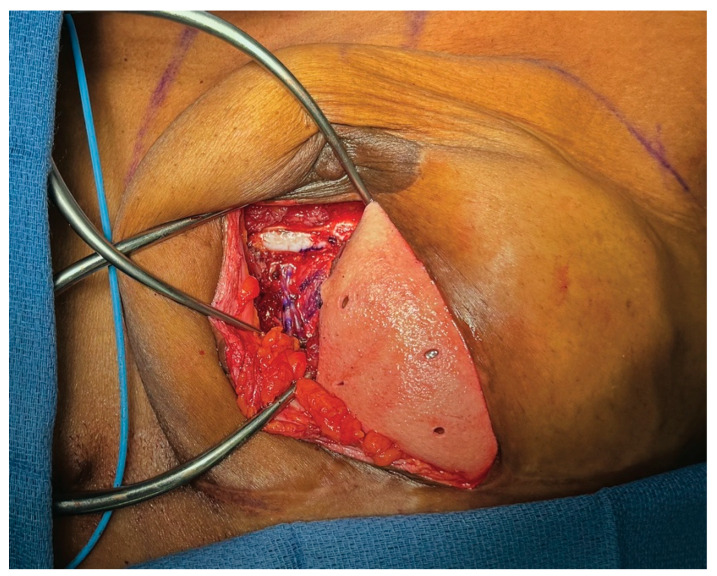
The left internal mammary vessels are exposed and prepared for anastomosis with the implant-ADM construct secured.

**Figure 7 medicina-61-01434-f007:**
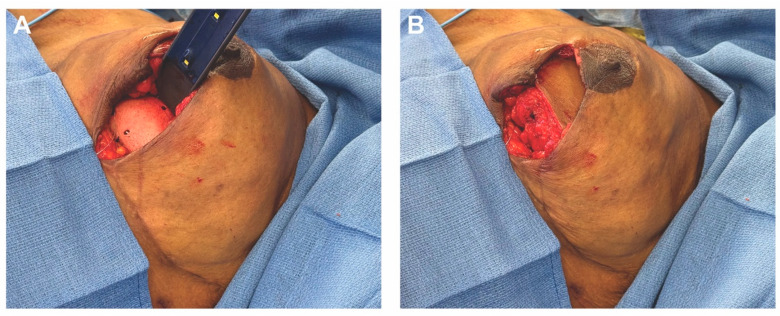
Inset of the autologous flap over the implant-ADM construct in situ. (**A**) The flap is inset over the implant-ADM construct using 3-0 PDS suture. (**B**) The flap provides complete coverage of the underlying implant-ADM construct.

**Figure 8 medicina-61-01434-f008:**
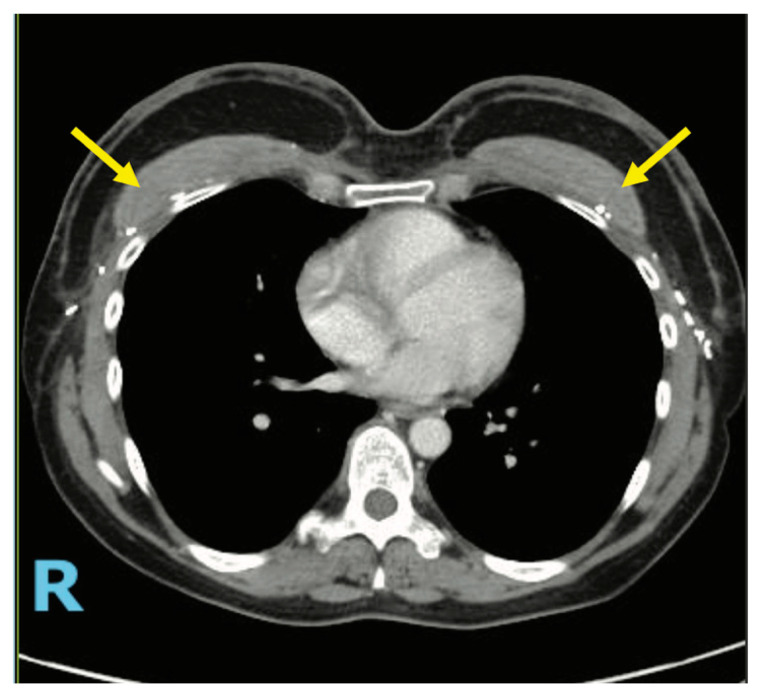
Axial CT image demonstrating the stacked ADM construct (yellow arrow) in a patient who previously underwent HyPAD^®^ breast reconstruction. The stacked ADM construct is clearly visible along the anterior chest wall, positioned deep to the autologous flap.

**Figure 9 medicina-61-01434-f009:**
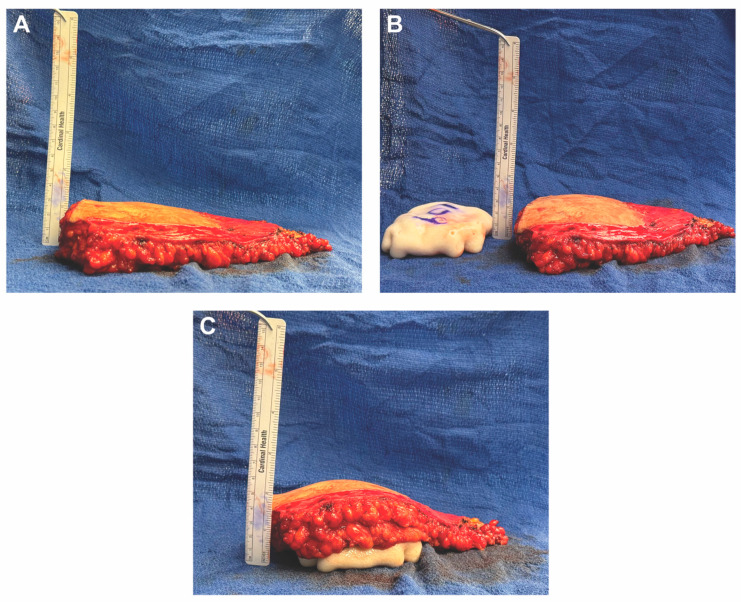
The stacked ADM construct provides additional volume and projection to the DIEP flap. (**A**) The DIEP flap alone had a projection of about 3 cm. (**B**) The stacked ADM construct alone had a projection of about 2 cm. (**C**) The DIEP flap combined with the stacked ADM construct had a projection of about 5 cm.

**Figure 10 medicina-61-01434-f010:**
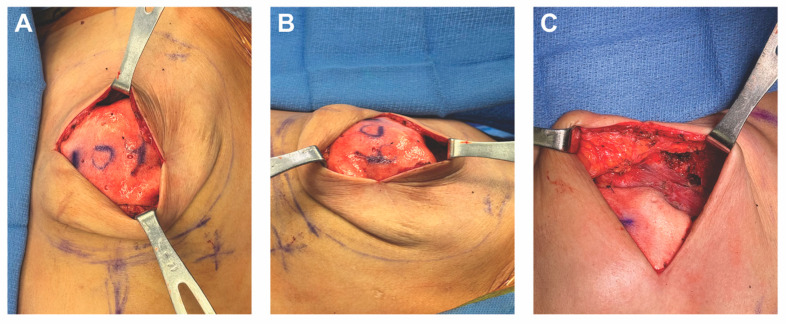
The stacked ADM construct is secured to the chest wall with a 3-0 PDS suture in an inferolateral position. (**A**) Frontal view of the stacked ADM construct. (**B**) Lateral view of the stacked ADM construct. (**C**) The construct is positioned to preserve access to the left internal mammary vessels superiorly and medially.

**Figure 11 medicina-61-01434-f011:**
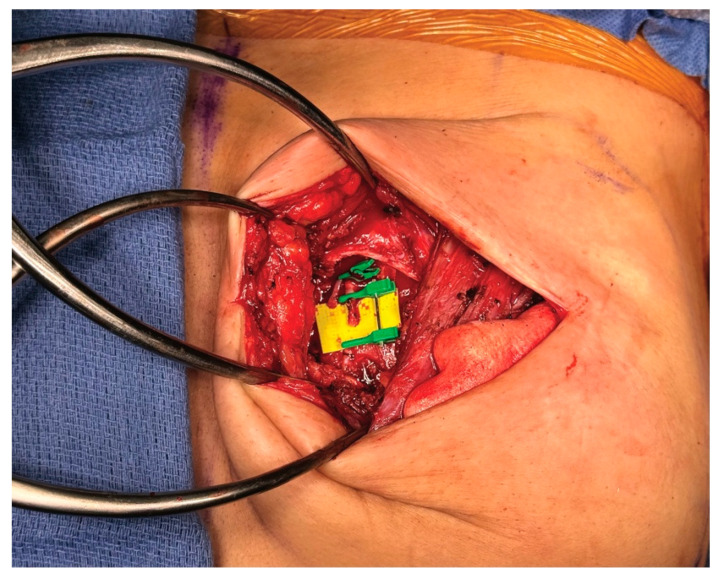
The inferolateral placement of the stacked ADM construct allows for unobstructed access to the left internal mammary vessels.

**Figure 12 medicina-61-01434-f012:**
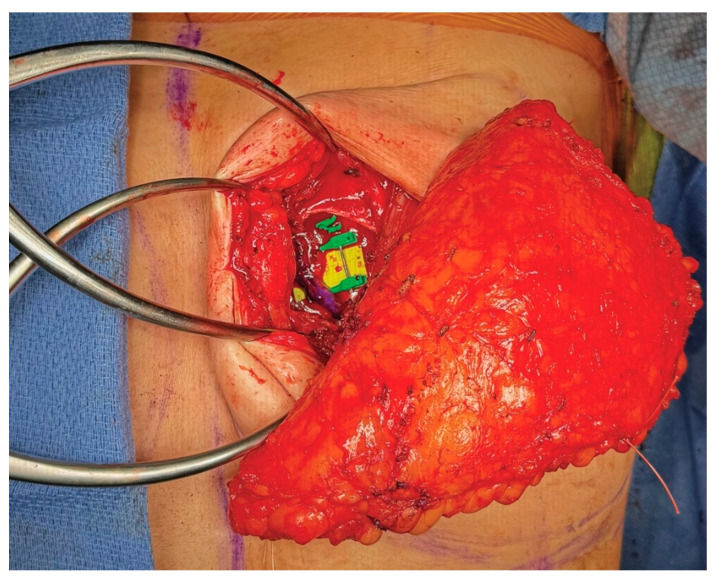
The autologous flap is inset over the previously placed stacked ADM construct, providing complete coverage. Microsurgical anastomosis to the left internal mammary vessels is then performed.

## Data Availability

No new data were created or analyzed in this study. Data sharing is not applicable to this article.

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
