# Peer review of "Revolutionizing Breast Reconstruction: The Rise of Hybrid Techniques"

_medicina, 2025, doi:10.3390/medicina61081434_

Round 1
Reviewer 1 Report
Comments and Suggestions for Authors
This is a well-structured narrative review addressing hybrid breast reconstruction (HBR), a novel surgical strategy combining autologous tissue flaps with prosthetic elements such as implants or ADM (acellular dermal matrix). The paper details two innovative approaches, HyFIL® and HyPAD®, including indications, surgical workflows, intraoperative steps, and patient outcomes. It also highlights clinical cases to illustrate technique feasibility.
1- The article introduces two clearly defined techniques that meet the unmet need of volume augmentation in patients with limited donor tissue.
2- The operative descriptions are highly detailed and well-illustrated, making the article valuable for reconstructive surgeons.
3- The authors strike a balance between depth and accessibility, offering both practical guidance and theoretical context.
Suggestion:
1- The manuscript would benefit from inclusion of numerical comparisons, e.g., complication rates, infection percentages, volume augmentation range, for HyFIL vs. HyPAD.
2- As a narrative review, the article does not qualify as a systematic evidence synthesis. This should be acknowledged more explicitly.
3- While complications are broadly discussed, the article should further differentiate risks specific to HyFIL versus HyPAD.
Conclusions: This is a timely and valuable contribution to the field of reconstructive breast surgery. Given the rise of hybrid approaches and the complexity of patient needs, this manuscript offers both innovation and guidance. After minor revisions, especially the addition of comparative data or summary tables, it would be highly suitable for publication.
Author Response
Suggestion 1: The manuscript would benefit from inclusion of numerical comparisons, e.g., complication rates, infection percentages, volume augmentation range, for HyFIL vs. HyPAD.
Response 1: Thank you for this recommendation. In response, we have incorporated numerical comparisons of volume augmentation capacity in the Discussion section (Page 3, Paragraph 3, Lines 347-348), specifying typical volume ranges for implants used in HyFIL versus ADM-only in HyPAD. While numerical data on long-term complication rates and infection percentages specific to each technique are not yet available, we acknowledge this gap and added a statement emphasizing that further research is essential to more precisely compare the risk profiles of HyFIL and HyPAD (Page 3, Paragraph 3, Lines 360-361). This addition is in a new paragraph within the Discussion section that differentiates the respective risks of the two techniques (see also Response 3). The revised manuscript now reads:
“Each technique described in this review presents its own set of practical considerations. In HyFIL, the use of implants requires patients to adhere to FDA-recommended screening and imaging protocols, increasing healthcare costs and time demands. Specifically, the FDA recommends MRI screening for silicone implants 5 years after placement and every 2 to 3 years afterward to detect silent rupture. These screenings, along with surgical interventions for implant rupture or expiration, contributed to over $33 million in healthcare costs in 2010 alone, with costs continuing to rise [45-47]. Patients hoping to avoid implants and their associated long-term surveillance demands may opt for the HyPAD technique. However, the augmentation capacity of even the thickest ADM constructs is limited to approximately 90 to 140 cc, whereas implants in this context range from 120 to 250 cc. As a result, HyPAD may be less suitable for patients requiring substantial additional volume. Currently, no technique provides substantial alloplastic volume supplementation without the use of implants. Furthermore, the use of ADM in breast reconstruction remains off-label under existing FDA guidelines.
The distinct risk profiles of these techniques raise additional considerations. Both techniques incorporate a flap and ADM, but only HyFIL uses an implant. As a result, HyFIL carries added risks associated with implant-based reconstruction, including infection, explant, malposition, capsular contracture, and unnatural breast contours [6]. These risks are absent in the HyPAD approach. Therefore, while both techniques share complication risks related to flap and ADM use, the presence of an implant in HyFIL introduces additional considerations for patient counseling, surgical planning, and procedural outcomes. Further research on long-term complication rates for HyFIL and HyPAD is essential to more precisely compare their respective risk profiles.”
Suggestion 2: As a narrative review, the article does not qualify as a systematic evidence synthesis. This should be acknowledged more explicitly.
Response 2: Thank you for this insight. We have added a Methods section to the manuscript (Page 3, Paragraph 1, Lines 88-94) specifying that this article is a narrative review of HyPAD and HyFIL breast reconstruction techniques, rather than a systematic evidence synthesis. The revised text now reads:
“Methods
This narrative review will explore two novel HBR techniques described by the senior authors (NT, MS): HyFIL (Hybrid Flap, Implant, Lipofilling) and HyPAD (Hybrid Flap, Prepectoral Acellular Dermal Matrix) [Figure 1] [28,36]. Specifically, the review will provide an overview of the clinical indications, operative workflows, surgical techniques, and patient outcomes associated with each HBR approach. Additionally, an illustrative clinical case performed by the senior authors (NT, MS) will be presented for each technique.”
Suggestion 3: While complications are broadly discussed, the article should further differentiate risks specific to HyFIL versus HyPAD.
Response 3: We appreciate this valuable suggestion. In response, we have added a paragraph to the Discussions section (Page 3, Paragraph 4, Lines 352-359) that explicitly differentiates between the risk profiles of HyFIL and HyPAD based on their distinct component structures. The revised manuscript now reads:
“The distinct risk profiles of these techniques raise additional considerations. Both techniques incorporate a flap and ADM, but only HyFIL utilizes an implant. As a result, HyFIL carries additional risks associated with implant-based reconstruction, including infection, explant, malposition, capsular contracture, and unnatural breast contours [6]. These risks are absent in the HyPAD approach. Therefore, while both techniques share complication risks related to flap and ADM use, the presence of an implant in HyFIL introduces unique considerations that may influence patient counseling, surgical planning, and procedural outcomes.

Reviewer 2 Report
Comments and Suggestions for Authors
This manuscript offers a well-written and up-to-date overview of hybrid breast reconstruction, focusing in particular on two techniques (HyFIL and HyPAD) developed by the authors to address volume mismatch in autologous breast reconstruction.
The article is clearly structured, didactic, and supported by a rich and relevant bibliography. The iconography is excellent, with high-quality intraoperative images and technical illustrations that significantly enhance the educational value of the paper. The clinical-practical approach is also commendable, with detailed descriptions of patient selection and surgical execution for each technique.
The only element that could be improved is the methodological structure of the manuscript. A dedicated Methods section is currently missing, which makes it difficult at first reading to understand whether the two clinical cases presented are original cases treated by the authors or derived from the literature. Including a brief methodological paragraph would clarify the type of work carried out and improve the overall transparency of the article.
Other than this, the manuscript is well executed and represents a valuable and informative contribution to the reconstructive literature.
Author Response
Suggestion 1: The only element that could be improved is the methodological structure of the manuscript. A dedicated Methods section is currently missing, which makes it difficult at first reading to understand whether the two clinical cases presented are original cases treated by the authors or derived from the literature. Including a brief methodological paragraph would clarify the type of work carried out and improve the overall transparency of the article.
Response 1: Thank you for this insightful recommendation. In response, we have added a Methods section to the manuscript (Page 3, Paragraph 1, Lines 88-94). This addition clarifies that the article is a narrative review on HyPAD and HyFIL breast reconstruction techniques. It also specifies that the two clinical cases included in the review were performed by the senior authors. The revised text now reads:
“Methods
This narrative review will explore two novel HBR techniques described by the senior authors (NT, MS): HyFIL (Hybrid Flap, Implant, Lipofilling) and HyPAD (Hybrid Flap, Prepectoral Acellular Dermal Matrix) [Figure 1] [28,36]. Specifically, the review will provide an overview of the clinical indications, operative workflows, surgical techniques, and patient outcomes associated with each HBR approach. Additionally, an illustrative clinical case performed by the senior authors (NT, MS) will be presented for each technique.”
